# Innate and Adaptive Immunity Linked to Recognition of Antigens Shared by Neural Crest-Derived Tumors

**DOI:** 10.3390/cancers12040840

**Published:** 2020-03-31

**Authors:** Giuseppe Donato, Ivan Presta, Biagio Arcidiacono, Marco F.M. Vismara, Annalidia Donato, Nastassia C. Garo, Natalia Malara

**Affiliations:** 1Department of Health Science, University Magna Graecia, 88100 Catanzaro, Italy; gdonato@unicz.it (G.D.); presta@unicz.it (I.P.); arcidiacono@unicz.it (B.A.); nastassiagaro@libero.it (N.C.G.); 2Department of Medical and Surgical Sciences, University Magna Graecia, 88100 Catanzaro, Italy; annalidia.donato@gmail.com; 3Department of Clinical and Experimental Medicine, University Magna Graecia, 88100 Catanzaro, Italy

**Keywords:** innate immunity, adaptive immunity, tumors neural crest-derived

## Abstract

In the adult, many embryologic processes can be co-opted by during cancer progression. The mechanisms of divisions, migration, and the ability to escape immunity recognition linked to specific embryo antigens are also expressed by malignant cells. In particular, cells derived from neural crests (NC) contribute to the development of multiple cell types including melanocytes, craniofacial cartilage, glia, neurons, peripheral and enteric nervous systems, and the adrenal medulla. This plastic performance is due to an accurate program of gene expression orchestrated with cellular/extracellular signals finalized to regulate long-distance migration, proliferation, differentiation, apoptosis, and survival. During neurulation, prior to initiating their migration, NC cells must undergo an epithelial–mesenchymal transition (EMT) in which they alter their actin cytoskeleton, lose their cell–cell junctions, apicobasal polarity, and acquire a motile phenotype. Similarly, during the development of the tumors derived from neural crests, comprising a heterogeneous group of neoplasms (Neural crest-derived tumors (NCDTs)), a group of genes responsible for the EMT pathway is activated. Here, retracing the molecular pathways performed by pluripotent cells at the boundary between neural and non-neural ectoderm in relation to the natural history of NCDT, points of contact or interposition are highlighted to better explain the intricate interplay between cancer cells and the innate and adaptive immune response.

## 1. Introduction

During embryonic development, between the 6th and 8th week of gestation, the cranial cells detach from the neuroepithelium of the dorsal neural tube, migrate along defined pathways to colonize the sites where they differentiate into various cell types. The variability of the fate of neural crest (NC) cells depends on the topographic disposition along the neural axis from which the cells, morphologically quite identical, initiated their migration. For example, cells migrating from the trunk region of neural tube will give rise to various sensory and autonomic neurons, associated glia, skin melanocytes, and chromaffin cells of the adrenal gland. On the other hand, the enteric innervation of the intestine originates from cells that have migrated from the vagal and sacral regions [1,2]. This explains how neural crest-derived tumors (NCDTs) form a heterogeneous group of neoplasms (Table 1). Neural crest cells (NCC) migration and tumorigenesis are both processes orchestrated by groups of genes whose modulation is responsible for the epithelial–mesenchymal transition (EMT) a process that confers to the cells’ new abilities like motility and invasiveness. The same molecular machinery engaged for EMT is discernible in the EMT characterizing the stemness profile of cancer cells of NCDTs. Antigens and cytokine secretion are largely shared between different NCDTs, regulating major mechanisms in tumor growth and progression, like intra-tumor inflammation [3,4].

The aim of this work is to resume and analyses the state of the art about the knowledge on tumorigenesis in NC derivatives. Also, we’ll try to underscore the common mechanisms working in embryonal development of these elements, their malignant transformation, the relationship with the immune system surveillance and possible targets for immunotherapy.

## 2. Neural Crest-Derived Cells (NCDCs)

The induction of NCDCs and neighboring cell types requires a highly orchestrated program of gene expression to regulate the long-distance migration, their proliferation rate, and pluripotency [5]. The pre-migrating phenotype of NCDCs has epithelial connotations with strong cell-cell contacts, apico-basal polarity and a specific conformation of actin cytoskeleton [6]. These cells develop extensive apico-basal cell surface polarity with actin microfilaments that are largely located at the cortex of the basolateral membranes; microtubules are predominantly baso-apically orientated influencing the position of cytoplasmic organelles, such as the Golgi apparatus positioned above the nucleus along the baso-apical axis (Figure 1) [7]. The junctional complexes are located on the lateral domain, and the basal domain interacts exclusively with the extracellular matrix through integrin receptors. The junctional complexes, particularly E-cadherin-associated adherent junctions, contribute to maintaining apico-basal polarity. The switch from pre-migratory in migratory NCDCs, is governed by two prevalent events: the acquisition of a mesenchymal moving phenotype and the reactivation of proliferation. The neural crest cell division rate is regulated both by density-related factors such as number of contacts with adjacent cells and by a combination of gradient models of bone morphogenetic proteins (BMP) with opposed inhibitory molecules such as chordin, follistatin, and noggin. Moreover, other genes such as those coding for dorsalin-1, zinc finger protein ZIC 2, wingless-type MMTV integration site family member (Wnt)-1 and Wnt-3a of secreted glycosylated proteins and growth factors such as basic fibroblast growth factor (bFGF) are involved [8]. Right during migration NCDCs undergo an extensive proliferation phase to generate a sufficient amount of precursor cells, needed to populate their target tissues. The NCDCs exodus starts synchronously in S-phase of cell cycle, promoted by the loss of epithelial cadherin with consequential and parallel reduction of cell–cell contacts during the EMT. The loss of epithelial cadherin follows an intracellular reprogramming of cadherin-expression that favors the synthesis of mesenchymal cadherin type (cadherin 7 and 11) and extracellular cleavage processes by the action of metalloprotease enzymes (MMP). The cleavage of epithelial cadherin releases the C-terminal fragments able, in turn, to promote the G1/S phase transition of cell cycle together with the cyclin D1 transcription [5]. These events suggest that proliferation and migration in NCDCs are phenomena strictly connected to an active epithelial–mesenchymal transition (EMT) pathway.

### 2.1. Epithelial-Mesenchymal Transitions in NCDCs

The EMT process, defined as phenotype change from epithelial to mesenchymal features, takes place in different embryonic territories and is mainly shared during the migration phase of NCDCs. At the beginning of migration, in neural crests, the oxygen gradient and the release of cell–cell adhesion plays a synergic central role. Hypoxia is able to activate HIF-1(hypoxia-inducible factor) α, which controls the expression of twist family BHLH transcription factor 1 (Twist-related protein 1) that, in turn, is able to repress expression E-cadherin to coordinate EMT and chemotaxis toward the chemokine SDF-1 by upregulating its receptor, C-X-C motif chemokine receptor 4 (CXCR4) [9,10]. This pathway is also involved in the homing of cancer cells into specific organs [11]. The EMT process is even linked to pathological contexts, both inflammatory and neoplastic, in which there is a remodeling of tissue matrix [12]. After migration reaching the tissue target, NSCDs reconvert their phenotype through the mesenchymal to epithelial transition (MET). The group of genes involved in EMT and MET are the same but undergo to an inversion of the sequences of their functional state. These genes are also mainly expressed by invasive cancer cells [5]. Rho GTPase family (RhoA, RhoB, and RhoC) signaling molecules act mainly through the cellular cytoskeleton and are important regulators of cell and tissue morphology and function. They are important mediators during diverse physiological processes like cell division, cell migration, wound healing, or immunity. RhoB can induce neural crest delamination by stimulating cytoskeletal remodeling and the assembly of focal adhesions between cells and extracellular matrix [13]. The Snail family of transcription factors is implicated in the differentiation of epithelial cells into mesenchymal cells during embryonic development, interfering in E-cadherin expression. Snail protein is expressed in invasive mouse and human carcinoma cell lines and tumors in which E-cadherin expression is lost [14]. Zinc-finger E-box binding homeobox (ZEB)-proteins are transcription factors bearing multiple functional domains and characterized by two widely separated clusters of C2H2-type zinc-finger domains with a central homeobox. Similarly, to snail 1 zinc finger protein (SNAIL), zinc finger e-box binding homeobox (ZEB)1 and ZEB2 transcription factors directly bind to the E-box elements and repress the transcription of the *CDH1* gene (coding for E-cadherin) promoting the EMT. ZEB2 drives development, delamination, migration, and specification of neural crest cells [15]. The suppression of *CDH1* transcription is also indirectly induced by Twist-related protein 1 and it is mediated by its transcriptional activation of snail family transcriptional repressor 2 (SLUG). In fact, SLUG knockdown blocks the function of TWIST to activate EMT in mammary cells [16].

Finally, signal transducer and activator of transcription 3 (STAT3) is another transcription factor involved in the neural crest specification and a downstream target of many neural crest signaling pathways. Overexpression of STAT3 promotes neural crest cell proliferation and its depletion. Instead, it inhibits the expression of genes important for neural crest development and proliferation. Furthermore, loss of function of STAT3 promotes apoptosis and loss of neural crest markers (e.g., SRY-Box Transcription Factor 10, SOX10 and Snail2). On the other hand, increased function promotes the maintenance of an undifferentiated neural crest phenotype [17,18]. EMT and inflammation are closely related. In fact, some of the genes aforementioned are involved in the regulation of the immune tolerance mechanisms of neural crest-derived tumors. In mouse melanoma xenografts, Snail1 transfectants that, in vitro, promote Treg expansion, can recruit Tregs and DCregs in vivo. The latter would depend on melanoma cell-derived C-C motif chemokine ligand 2 (CCL2) [19].

Moreover, intratumoral injection of an anti-Snail1 siRNA can decrease local immunosuppression and resistance to DC immunotherapy, underlining the central role of EMT in cancer [20] regression in this model.

### 2.2. NCDCs and the Immune System

Relatively few is known about the role of the NCDCs on the immune system regulation during the embryonic phase. A certain role of the NCDCs was defined in the embryologic development of the thymus. In fact, the ablation of small portions of neural crest alters the development of the thymus [21]. These observations establish, experimentally, a presumptive role of neural crest in the development of the immune system. Moreover, these observations are supported by the downstream effects registered in the neurocristopathies or other disorders compromising the normal function of NCDCs, such as the Hirschsprung’s disease or Di George syndrome and in the recent infection of Zika virus.

The Hirschsprung’s disease (or colonic aganglionosis) the typical signal signals, neurotrophic factor (GDNF)-family ligands transduced by Ret and GFRa1 receptors and on endothelin 3 (ET3)/endothelin receptor B (ETRB) pathway, are absent. Both GDNF and ET3 are produced by the gut mesenchyme where NCDCs migrate and differentiate. GDNF provides survival/ differentiation and chemotactic signals to ret-expressing migratory NCDCs [22], whereas ET3 inhibits premature NCDC neuronal differentiation, resulting in incomplete colonization of the bowel [23]. The pathogenesis of this nerocristopathy results in a defective craniocaudal migration of neuroblasts originating from the neural crest and differentiation of neuroblasts into ganglion cells [24]. The resulting follicular architecture of the lymph nodes located along the anti-mesenteric surface of the bowel is altered compromising the maturation of the B-lymphocytes and favoring the maturation of plasma cells with an increase in IgA-containing plasma cells [22]. GFRA1 has also been implicated in cancer cell progression and metastasis. Recent findings show that GFRA1 contributes to the development of chemoresistance in osteosarcoma. Ret proto-oncogene germline mutations are crucial for the onset and the progression of medullary thyroid tumors, and the occurrence of single nucleotide polymorphisms could predispose to the sporadic forms [23]. The Di George syndrome (DGS) is a congenital disease resulting from defective development of neural crest cells that colonize the pharyngeal arches and contribute to lower jaw, neck and heart tissues [25]. DGS causes cardiac outflow tract anomalies, craniofacial dysmorphogenesis, thymus hypoplasia, and mental disorders. The main gene underlying the pathogenesis of these defects in human patients and mouse model is the SDF1/CXCR4 axis. The SDF1/CXCR4 is involved in the chemotactic guidance and impaired in cortical interneurons of mouse DGS models [25]. CXCR4 is involved in the regulation of many immune system functions: chemotaxis, proliferation, apoptosis, survival, and differentiation. Differential expression of CXCR4 is critical for the maturation and selection of B cells [26,27] and sustained by hypoxic gradient. The SDF1/CXCR4 is involved in the inflammation surrounding metastatic progression of many carcinomas [28] comprising the NCDTs like in schwannoma [29].

Zika virus (ZIKV) infection during pregnancy is linked to microcephaly, which is attributed to infection of NCDCs. ZIKV-infected NCDCs are characterized by limited apoptosis and are induced to produce cytokines that promote embryopathies through signaling crosstalk [30]. Experimental studies registered an increase in secretion of LIF, VEGF, IL-6, and other molecules by ZIKV-infected NCDCs that could have both autocrine as well as paracrine effects on the developing of central nervous system. The defects on the maturation of the neural crests, direct or indirect, reported in the genetic pathologies described above, result in a common alteration involving the production of cytokines and resulting in the impairment of B-lymphocyte maturation and altered T-lymphocyte cytotoxicity. Thus, NCDCs and relative tissues could exhibit immunomodulation properties. Many tumors commonly secrete these cytokines. Interestingly, some NCDTs under stress begin the production of cytokines as reported recently in human melanoma cell lines under influence of norepinephrine become able to influence tumor progression by modulating the expression of VEGF, IL-8, and IL-6 [31].

Other cytokines released by NCDCs are stem cell factor (SCF) and ET3 that activate, respectively, c-kit TKR and ETRB/B2 at the NCC surface [32]. The migration and survival responses to SCF have been analyzed in mutant mice [33,34]. The physiological mechanism of ET3 induction in the adult is SCF-KIT-dependent, and it was described in melanocyte lineage differentiation in the skin and recently in human umbilical vein endothelial cells and melanoma cells in vitro, gastrointestinal stromal tumors, human sun-exposed skin, and myenteric plexus of human colon post-fasting in vivo [32].

Moesin-ezrin-radixin-like protein (Merlin), also known as schwannomin is a tumor suppressor protein encoded by the neurofibromatosis type 2 gene NF2. Loss of function mutations or deletions in NF2 cause Neurofibromatosis type 2 (NF2), a multiple tumor-forming disease of the nervous system. Interestingly NF2 mutations and merlin inactivation, enhancing the NFKB’s transcriptome with consequential increase and release of pro-inflammatory cytokines, occur in spontaneous schwannomas and meningiomas, as well as other types of cancer including mesothelioma, glioma multiforme, breast, colorectal, skin, clear cell renal cell carcinoma, hepatic and prostate cancer [29].

Multiple roles of TGF-b and BMP signaling molecules Bone morphogenetic proteins (BMPs) and transforming growth factor-b (TGF-b) are involved in multiple stages of neural crest development. In the early embryo, BMPs influence commitment of the dorsal neural tube into NC [35]. At later stages, BMPs are necessary and sufficient for sympathetic neuron development and several BMP downstream genes were characterized. The role for BMP2, BMP4, and TGF-b in driving NCSCs to sympathetic-like neuronal fate was demonstrated in vitro [36]. BMP4 stimulates meningioma cell proliferation and phosphorylation/activation of Smad1 playing autocrine/paracrine roles and interacting with other transforming growth factor-beta superfamily members in regulating meningioma growth and differentiation.

Equally, little it is known about potential regulatory influence on the immune system by normal mesenchymal stem cells, neural crest-derived, during adult life. Research has described the immunosuppressive properties in vitro of mesenchymal stem cells (MSCs) resident in some neural crest-derived tissues: corneal endothelium, gingivae, and oral mucosal progenitor cells [37]. In particular, the endothelial cells of corneal and MSC resident in the dental pulp, express PD-L1 (B7-H1) and TGF-β and are able to inhibit T cell proliferation and to induce regulatory T cells [38,39]. PD-1 is a key surface molecule promoting self-tolerance by suppressing T cell inflammatory activity and controlling the maintenance of the stem-cell phenotype [39]. This immunosuppressive property is preserved in many NCDTs in the advanced phase characterized by an increase expression of PD-1 [40,41,42,43,44].

## 3. Neural Crest-Derived Tumors (NCDTs)

### 3.1. Melanocytic Tumors Cutaneous/Mucosal

Malignant melanoma arises from cells known as melanocytes and it is the most aggressive kind of skin cancer with a constantly growing incidence [45]. There is a growing body of evidence to suggest that mutations in neuroblastoma ras viral oncogene homolog (N-RAS) and v-raf murine sarcoma viral oncogene homolog B (BRAF), both activators of mitogen-activated protein kinase (MAPK-ERK), are involved in melanoma development [46,47]. In melanoma, RhoB deficiency causes hypersensitivity to BRAF and MEK inhibitors-induced apoptosis [48]. In addition, oncogenic signaling, mutations, proto and repressor genes have been demonstrated to be associated with melanoma progression and its EMT phenotype [49,50]. Murtas and colleagues investigated the association between N-cadherin and Notch1 in primary cutaneous melanomas, showing that a concomitant high expression of N-cadherin and Nothc1 previses an adverse prognosis in patients with melanoma [51]. A great number of studies demonstrated that an abnormal activation of phosphatidylinositol-4,5-bisphosphate 3-kinase /Akt-protein kinase B (PI3K/Akt) signaling axis is connected with EMT, since it causes a strong expression of N-cadherin and vimentin genes [52]. In addition, the Wnt pathway, once activated, induces the expression of N-cadherin, and Vimentin, while, on the other side, it decreases the expression of cyclin D1, c-Myc, and p-GSK3β [53]. Moreover, Wnt signaling pathway has been shown to possess a similar ability to promote melanoma progression and metastasis [54]. Shi and colleagues demonstrated that there was an inverse correlation between miRNA22 and its target gene FMNL2 in melanoma patients, modulating EMT and Wnt/β-catenin [55]. In another work, Zhu and colleagues showed that the over-expression of miR-3662 halted the EMT program, affecting the expression of matrix metalloproteinases 3 and 9. In addition, these authors demonstrated that miR-3663 inhibited EMT process by targeting the transcription factor ZEB1 (proto-oncogene zinc finger E-box binding homeobox 1) and therefore blocking the transition of epithelial cells to a mesenchymal condition [56]. The EMT phenotype is common in cancer stem melanoma cell characterized by a positivity for CD133 and PD1 and by CSC intracellular markers comprise enzymes such as aldehyde dehydrogenases (ALDH), and transcription factors such as Sox2 and Klf4.

There is growing evidence that the production of cytokines in melanoma is supported by stromal cells or by melanocytes themselves and may constitute a further process in tumor survival and growth. In this regard, Barbero and colleagues demonstrated that Wnt5a signaling forced melanoma cells to produce and secrete IL-6, IL-11, IL-8, IL6 soluble receptor, TNF soluble receptor I and MCP-1, that were, in turn, responsible for angiogenesis, cell proliferation, exhaustion of effector T cells, pharmacological resistance, and cancer cell survival. Furthermore, the pro-inflammatory condition and Wnt5a signaling sustained autocrine positive feedback loops that included NF-kB, IL-6, STAT3 and other immune cells, causing an immunomodulatory effect on melanoma and conferred survival and therapeutic resistance [57,58]. Kholmanskikh and colleagues focused also their attention on interleukins secretion. They reported that melanoma cell lines expressed the IL-1 receptor and that its activation via NF-kB and JNK pathways caused a downregulation of microphthalmia-associated transcription factor (MITF-M), helping melanoma cells to resist elimination by T lymphocytes [59,60]. The family of transcription factor called nuclear factor-kappa B NF-kB pathway is mainly associated with the activation of hypoxia-inducible factors (HIFs) involved in a demonstrated crosstalk in hypoxia and inflammation. These two main molecular players have been demonstrated to be over-expressed in the melanoma cells [61].

### 3.2. Medullary Thyroid Carcinoma

Medullary thyroid carcinoma (MTC) originates from the parafollicular C cells of the thyroid that secrete calcitonin and accounts for about 5% of all thyroid cancers [62]. MTC has an important genetic component with about a quarter of all MTC being hereditary (familial medullary thyroid carcinoma or FMTC) and carrying a mutation in the RET protooncogene [63,64]. The FMTC constitutes a variant of the Multiple Endocrine Neoplasia type 2A (MEN2A) phenotype, characterized together with MEN2B variant phenotype, by an increased risk to develop pheochromocytoma [65].

From the clinical point of view the occurrence of metastatic spread of MTC is the crucial event that drastically impoverishes the prognosis. The acquisition of this feature, from a diverse range of malignancies, has been associated to the expression of the G protein-coupled chemokine receptor CXCR4 and its ligand CXCL12 [66]. The CXCR4 receptor, normally expressed in diverse tissues during embryonic development, is almost absent in adult tissues in physiological conditions but then reactivate its expression in many types of carcinomas. A mechanistic feedforward loop was identified for CXCR4 and HIF-1α subcellular expression levels. Nuclear CXCR4 promoting HIF-1α accumulation, which, in turn, induced CXCR4 transcription, are associated with poor prognosis independently and when combined with TNM stage [67].

The chemokine CXCL12 functions as ligand for CXCR7 too and this other receptor routes on signals that inhibit apoptosis and cooperate to adhesion properties of tumor cells [66,68]. Even in MTC, it was found that the expression of CXCR4 correlates with advanced tumor stage and metastatic phenotype while that of CXCR7 declined in metastases. In the same study, it was shown that rh-CXCL12 and also CXCR4 agonists, pushed MTC cells to enter the G2/M cell cycle phase in a CXCR4-mediated manner and induced the expression of genes associated with EMT and tumor cell invasion. It was found an increase of the expression of Bone Marrow Stromal Cell Antigen 2 (BST2 gene), Fibroblast Growth Factor 9 (FGF9 gene) and Vimentin, while E-cadherin levels decreased [69]. The cancer stem cells in medullary thyroid carcinoma are characterized by chemo- and radio-resistance induced by increased signaling pathways, such as signal transducer and activator of transcription 3 (STAT3), c-Met, SOX2, rearranged during transfection (RET), CD44. Moreover, recently it is demonstrated an immunosuppressive action by cancer stem cells of medullary thyroid carcinoma, through the expression of PD-1 [70] that justified the beginning of clinical trial with the Pembrolizumab.

### 3.3. Pheocromocytoma and Paragangliomas

In many tissues, there are dispersed cells producing and secreting bioactive substances, which can be distinguished from the endocrine glands. These cells have hybrid characteristics between epithelial cells and neurons and constitute the so-called dispersed neuroendocrine system (DNS) [71]. For these cells, one of these two profiles, epithelial or neuronal may prevail over the other. Paraganglia (PGs) are small agglomerations of neuroendocrine cells spread throughout the body from the base of the skull to the coccyx. The tumors that originate from neuroendocrine cells with a more pronounced neuroectodermal phenotype are divided into two groups: paragangliomas and not paragangliomas. Paraganglia are groups of cells of neuroectodermal origin that produces catecholamines and regulatory peptides. Gangliocytic paragangliomas are to be considered atypical paragangliomas with very large cells, similar to gangliocytes, and the matrix similar to schwannomas together with the typical characteristics of paragangliomas (spheres of “zellballen” cells) [72]. Pheochromocytoma and paraganglioma are metastatic in 15% of cases. Pheochromocytoma susceptibility may be associated with germline mutations in the tumor-suppressor genes *VHL* and *NF1* and in the proto-oncogene *RET*, and recently has been associated with germline SDHD mutations. *VHL* gene product has a critical role in the regulation of hypoxia-responsive genes. Germline mutations in the tumor-suppressor gene VHL, both in human cancers and in mouse models, resulting in upregulation of angiogenic growth factors, such as vascular endothelial growth factor, secondary to a defect in ubiquitylation of the α-subunits of the HIF-1 and the HIF-2 transcription factors [73]. The SDHB gene, coding for the succinate dehydrogenase complex iron sulphur subunit B, is one of the most frequently mutated genes in these malignancies. The presence of SDHB mutation has been associated with the induction of expression of genes responsible for endothelial mesenchymal transition like LOXL2, TWIST, SNAI1, TCF3, MMP2, and MMP1. While genes typical of epithelial phenotype like KRT19 and CDH2 were downregulated [74,75] and ZEB1/δEF1, and Slug are induced during progression of metastatic Pheochromocytoma [76]. 

Moreover, pheochromocytoma and paraganglioma over express PD-1 and for this reason, are good candidates for immunotherapy [77]

### 3.4. Neuroblastoma

Neuroblastic tumors (NTs) are the commonest extracranial solid malignancy of childhood and are embryonal tumors that arise from the sympatoadrenal neuroendocrine system and in the 50% of cases of NT arise in the adrenal medulla. Neuroblastoma (NBs) is characterized by small tumor cells with hyperchromatic nuclei and scanty cytoplasm, it is more or less differentiated and aggressive. The amplification of the N-MYC proto-oncogene is present in about 30% of NBs and is associated with a more aggressive course. TGF-β1 can trigger EMT in neuroblastoma. In fact, E-cadherin significantly decreases, whereas α-SMA significantly increases after neuroblastoma cells are treated with different concentrations of TGF-β1 [78] and increase Slug and Zeb-1 expression [79]. Both N-Myc and c-Myc proteins in NB can bind and activate the *TWIST1* promoter. Therefore, *TWIST1* may be a direct MYC transcriptional target [80]. Both *TWIST1* and TGF-β1 are in turn, regulate by directly or indirectly binding to the hypoxia-response element (HRE) HIF-dependent [81].

The noncanonical Wnt-planar cell polarity (PCP) signaling cascade is fundamental for the migration of neural crest cells. PCP proteins control the activity of Rho GTPases by an activation or an inhibition of RhoA and Rac1. The stimulation of Rho signaling by PCP results in downstream activation of the serine/threonine Rho-associated coiled coil-containing protein kinases (ROCK)1 and ROCK2. ROCK may be a promising target for the treatment of high-risk neuroblastoma patients expressing high MYCN levels. In fact, genes controlling the activity of ROCKs are mutated in approximately 30% of neuroblastoma and high ROCK2 expression in neuroblastoma tumors corresponds to poor patient survival [82].

Tumor-Associated Macrophages (TAM) are strictly associated with poor survival in neuroblastomas that lack MYCN amplification. Amplification of MYCN or constitutive upregulation of MYC protein is observed in approximately half of high-risk tumors; TAMs play a role as inducers of MYC expression in neuroblastomas lacking independent oncogene activation. In fact, MYC upregulation correlates with markers of TAM infiltration such as IL-6-that activate STAT3 in this mechanism [83]. Moreover, an additional mechanism of escape by neuroblastoma of the immune checkpoints, inhibitory pathways that physiologically maintain self-tolerance and limit the duration and amplitude of the immune responses is represented by the PD-1/PD-Ls axis [40].

### 3.5. Schwannoma, Neurofibroma, and MPNST

Is a benign, typically encapsulated nerve sheath tumor composed entirely of well-differentiated Schwannn cells, with loss of merlin (the NF2 gene product) expression in conventional forms. Usually schwannomas (SC) are sporadic and solitary neoplasms, but multiple schwannomas are associated with neurofibromatosis type 2 (NF2) and schwannomatosis [84]. Immunohistochemistry of Sc shows a phenotype with clear expression of S100 protein and transcription factor SOX10 [84]. Because the relevant clinic impact and the difficulties in therapeutic approach the majority of studies is dedicated to vestibular SC. Moreover, in human vestibular SC, immunohistochemical examination reveals an increasing of pro-inflammatory cytokines, including TGF-β1, IL-1β, and IL-6 that may promote growth of the neoplasm [85] supported by the loss of Merlin protein with resulting enhance of NFKB transcriptome [29]. Macrophage colony-stimulating factor (MCS-F) and IL-34 are also associated with growth of vestibular schwannoma, probably by activation of TAMS [86]. Inactivating mutations of the tumor suppressor gene NF2 that encode the protein merlin have been observed in 60% of all SCs [84]. In response to different stimuli, macrophages may undergo classical M1 activation state (stimulated by TLR ligands and IFN-γ) or alternative M2 activation state stimulated by IL-4/IL-13). Ml macrophages can produce high levels of inducible nitric oxide synthase (iNOS) to promote arginine metabolism into nitric oxide and citrulline and high levels of proinflammatory cytokines. M2 macrophages express scavenger receptors, mannose receptors and IL-10 and are involved in tissue remodelling, immune modulation and tumor progression [87]. Interestingly, M2-type macrophages in vestibular schwannomas relate to angiogenesis and volumetric tumor growth [88]. However, in recurrent SCs after partial resection and radiotherapy, the dominant population of TAMs is represented by M1-activated elements [89]. From an embryological point of view, it is very interesting the entity called melanotic schwannoma (MS). Such an aggressive neoplasm shows all the characteristics of benign schwannomas together with pigmentation, melanosomes, and expression of immunohistochemical markers of melanomas. Moreover, the histopathological pattern of MS may contain or not calcification type psammomatous bodies. Tumors containing psammomatous bodies are present in 50% patients, together with a Carney complex, and involve autonomic nerves of viscera, such as the intestinal tract and heart. This association may be a further clue of an origin of cardiac myxomas from neural crests (see below) [84]. Sporadic neurofibromas are common cutaneous tumors. Instead, multiple and plexiform neurofibromas are associated with type 1 neurofibromatosis (NF1) [89]. S100 protein and transcription factor SOX10 are immunohistochemically expressed in a lesser extent in NFBs than in Sc. Intermingled perineurial cells show a positivity for epithelial membrane antigen (EMA). The NF1 is caused by mutations of the NF1 gene on chromosome 17q11.2, encoding neurofibromin which harbours a GAP-related domain, and belongs to the group of mammalian RAS-GAPs. Deregulation of RAS signaling in NF1 results in the development of multiple neurofibromas. Neurofibromin plays a crucial role in suppressing the expression of EMT-related transcription factors and signaling associated with EMT may contribute to neurofibroma formation in NF1 patients. In fact, depletion of neurofibromin in normal human Schwann cells promotes the expression of EMT-related transcription factors. In addition, such EMT-related transcription factors are up-regulated in cultured NF1 Schwann cells and NF1-associated neurofibroma specimens. Immunohistochemical analysis of neurofibromas show a marked expression of ZEB1 [90]. In NFBs, neoplastic Schwann cells produce factors such as chemoattractant stem cell factor that recruit Mast cells and Macrophages. Macrophages in neurofibromas stronger pro-inflammatory M1 signatures than M2. This finding suggests the presence of a continuous inflammation in the neurofibroma microenvironment [91].

Malignant peripheral nerve sheath tumor (MPNST) is a malignant tumor with evidence of Schwann cell or perineurial cell differentiation, commonly arising in a peripheral nerve or in extraneural soft tissue [84]. MPNST is a rare, highly aggressive sarcoma, which can occur in one of three clinical scenarios: (1) in the context of NF1, (2) following radiation therapy, and (3) sporadically. By immunohistochemistry, MPNST is at least focally positive for S100 protein and SOX10 in fewer than 50% of cases. GAP43, a molecule essential in neural development and axonal regeneration, are also positive in MPNST and other nerve sheath tumors [92].

All types of MPNST show highly recurrent genetic inactivation in NF1, CDKN2A, and the PRC2 components SUZ12 and EED. PRC2 consists of the proteins EED, SUZ12, and EZH1/2, and, among other roles maintains a crucial balance of the di- and tri-methylation of the H3K27 histone [84,92]. Interestingly, a subpopulation of cancer stem cells positive for CD133 is identifiable in human primary malignant peripheral nerve sheath tumor (MPNST). Such cells show enhanced invasiveness which is linked to the increased expression of β-Catenin and Snail, two important proteins involved in EMT.

From an immunological point of view, MPNST is characterized by low PD-L1 and absent PD-1 expression with significant CD8^+^ tumor-infiltrating lymphocytes (CD8^+^ TIL) presence. These data suggest that immunotherapies may offer some benefit for MPNST. Whereas, MPNST immune microenvironment does not correlate with patient outcome [44,93]. As we have shown, Sox10 and S100 are recognized by pathologists as markers of neural crest origin of tumors. However, S100 protein is expressed also in many other types of cells and tumors of various origins. For such a reason the most specific marker for benign and malignant peripheral nerve sheath tumors in most cases is Sox10 [94,95].

### 3.6. Meningioma

Meningioma is defined as a group of mostly benign, slow-growing neoplasms that most likely derive from the meningothelial cells of the arachnoid layer. Meningiomas are classified into three major groups according to their WHO grade and biological behaviour [84,96].

The first genetic alteration observed in meningiomas was monosomy of chromosome 22 or deletion of 22q; the key gene involved, NF2 is the gene involved in the genesis of the type-2 neurofibromatosis, and is located on 22q12, encoding the tumor suppressor merlin [97,98]. Interestingly, genetic abnormalities in meningiomas are different in relation to the location and the type of the lesion: in fact, SMO and AKT1-MTOR mutations are quite common in non-NF2, genomically stable meningiomas of the skull base. On the other hand, meningiomas with NF2 inactivation are genomically less stable and localize to the brain hemispheres. Moreover, mutations of NF2 are most frequent in fibroblastic/transitional meningiomas, KLF4 and TRAF7 in secretory meningiomas, and AKT1 mutations in grade I meningothelial meningiomas of the skull base and spine [97]. 80% of patients with recurrent chordoid meningiomas have complete deletions on all four chromosomal loci 22q, 18p, 14q, and 1p [99,100]. Analyzing 85 meningiomas of various grade and type, were observed reduced expression of EMT-linked transcription factors such as Twist, Zeb-1 and Slug. Interestingly, in recurrent compared to non-recurrent meningiomas reduced E-Cadherin and increased Slug levels were observed. Moreover, comparing aggressive grade II or III meningiomas with grade I tumors, loss of Zo-1 and E-Cadherin together with increased expression of Zeb-1 and Slug were observed, indicating that molecular features of EMT are present in aggressive meningiomas [101]. Immunohistochemical profile of meningiomas suggests an intermediate state between EMT and its reversal state: mesenchymal-epithelial transition (MET). Indeed, epithelial membrane antigen (EMA) and vimentin are typical immunophenotypic markers of meningiomas. Both innate and adaptive immunity play a role in the genesis and progression of meningiomas. The so-called lymphoplasmacite-rich meningioma is a variant that show a heavy inflammatory infiltrate. Because macrophages often predominate and plasma cells are not always conspicuous, the alternative name “inflammatory-rich meningioma” has been proposed. Interestingly, systemic haematological troubles such as iron-refractory anemia and hypergammaglobulinemia have been observed [84]. Grade II meningioma called chordoid meningioma is a neoplasm with morphological features resembling chordoma. Such a tumor can be associated to an inflammatory Castleman-like syndrome, due to a secretion of interleukin-6 (IL-6). Chordoid meningioma is rich of cells of innate immunity such as macrophages and mast cells [102].

Sometimes in meningiomas infiltration by tumor-associated macrophages (TAMs) may be very pronounced, especially in cases of meningiomas that carry isolated monosomy 22, where the immune infiltrates also contain greater numbers of cytotoxic T and NK-cells that are associated with an increased anti-tumoral immune response. In line with this, the presence of regulatory T cells is usually limited to a small fraction of all meningiomas [87]. These studies reflect an attempted immune response. However, the expression of PDL1 by meningioma cells and its potential role in local immunosuppression was recently established [43]. Such a variety of inflammatory response to tumor meningioma tissue suggest that a traffic of cytokines has a pivotal role in the development of such neoplasms. VEGF is secreted by meningothelial cell, but is often produced by TAMs in cancer tissues, VEGF secretion correlate with intratumoral angiogenesis a peritumoral edema [103]. TGF-beta inhibition of growth in normal leptomeninges can be lost in meningiomas. Moreover, loss of TGF-beta and bone morphogenetic protein signaling components and TGF-b type III receptor may contribute to the development and progression of higher grade meningiomas [104]. Moreover, all meningiomas, which progressed from low- to high-grade meningiomas, were HIF-1 positive [105].

### 3.7. Tumors Of Not Established NC Origin

A separate section, including all tumors of uncertain histogenesis but probably deriving from neural crest cells can be set. Cardiac myxomas (CMs) are rare primary cardiac neoplasms, which share many characteristics with the derivatives of neural crests [106]. Usually sporadic and single, CMs arise from the foramen ovale region, in the left atrium. When in association with the Carney complex (CC), CMs can occur in younger patients, at multiple sites of the heart. Cardiac neural crests are a segment of neural crests that contribute to build the outflow tract and cardiac septum. Moreover, cephalic NCCs are the source of vascular smooth muscle cells and pericytes in an area of the circulatory system that extends distally from the cardiac outflow tract, along arteries and jugular veins, to the distal capillary beds and choroid plexuses of the face and forebrain. CMs express many markers common in neural crest such as calretinin, Sox or S100 and PGP 9.5. Its EMT is characterized by increase expression of N cadherin and Notch [107]. Some cardiac myxomas are characterized by Il-6 secretion corresponding to a clinical syndrome like Castleman disease. Moreover, the cardiac mixoma is characterized by infiltration of macrophages and mastocytes [108,109]. Sometimes, after surgical removal, myxomas re-emerge causing dysfunctional heart. This malignant behavior is denounced by the expression analysis of c-MYC, HIF-1α, p53, and vimentin considered potential biomarkers for malignancy detection in myxoma [110]. Finally, up to 10% of CMs occur in the Carney Complex (CC) context, which is familial in 70% of cases with autosomal dominant inheritance. Germline inactivating mutations of the PRKAR1A gene are found in 37% of patients with sporadic CC and in more than 70% of patients with familial CC. Patients with CC are prone to develop multiple schwannomas though, more peculiarly, they develop psammomatous melanotic schwannomas [106]. Lung cancer is the leading cause of cancer-related mortality around the world, but many problems concerning its therapy and diagnosis are at present unsolved [111].

Small cell lung cancer (SCLC) derive from bronchial Kulchitsky cells and is a high malignancy neuroendocrine tumor. It represents 20% of all lung tumors and in 70% of patients, it is diagnosed when it is already at an advanced stage. Neuroendocrine cells are probably derived from different embryonic leaflets. The enteroendocrine cells would probably derive from the entoderm, whereas those of the lung, larynx and thymus would be more likely to originate from the neural crest. Neural-like features are not uniformly distributed throughout the diffuse neuroendocrine system. Thus, the neuroendocrine cells located in the larynx, lung, thymus, and thyroid (C cells) are the most ‘neural-like’ cells of the system, a feature that becomes obvious in the corresponding tumors [112]. Many features confirm the origin of SCLC from neural crests: S100, CD56 (N-CAM) and synaptophysin are common markers expressed in small cell lung cancer. Interestingly, small cell carcinoma is often responsible for paraneoplastic syndromes, among which, the possibility of ectopic calcitonin secretion is found, as it happens in medullary carcinoma of the thyroid [113]. The initiating molecular events of SCLC are believed to be inactivation of TP53 and RB1, which mainly occurs in Kulchitsky cells in the respiratory epithelium, although many other genes and signaling pathways are disrupted, especially Notch signaling [114]. Notch signaling play various roles during neural crest development, especially in neurons differentiation [115,116]. Interestingly, TAMs are probably involved in SCLC progression through STAT3 activation and TAM-derived IL-6 can be one of molecules related to STAT3 activation in SCLC cells [117]. STAT3 is involved in the neural crest specification and a downstream target of many neural crest signaling pathways (see above). The expression of PD-1 in small cell lung cancer was previously described [118].

## 4. Conclusions

The NCDTs pattern appears characterized by divergent specificities, and at the same time, maintains common features. The discrepancies originate probably from the multipotency of the neural crest considered, for this reason, as a fourth embryonic leaflet. The similarities founded, on the other hand, appear a consequence of the EMT pathways early activated during embryonic life. In other words, the EMT could be responsible to influence the reactivation of common pathways and the re-expression of common antigenicity shared by both normal and in cancer stem cells. On the other hand, the MET process could be understanding the different features characterizing NCDTs. The immunosuppressive activity of normal progenitors, through the expression of PD-1 and the production of cytokines involved in inhibiting the cytotoxic activity of T cells, is common to many of the NCDTs tumors. Certainly, the stem cancer cell makes use of reactivating an immunosuppressive pathway, partly preserved in the adult stem cells, but necessary, together with other similar immunological suppression systems, during embryonic life, to guarantee maternal-fetal tolerance and inducing an intraembryonic tissue immune tolerance. Multiple advantages are acquired by cancer cells by activating immunosuppressive pathway, for example, reduced aggression by the cell-mediated immunity, preservation of stemness features and of EMT profile to guarantee proliferation and migration favoring the disease progression (Figure 2). Immunotherapy that interferes with the immunosuppressive activity of cancer stem cells, through PD1 or other pathways, it acquires the potency to revert the multiple benefits trigger by cancer cells of tumor neural crest-derived and ensures a good control of disease. Finally, the point of view proposed in this review suggest that, could be useful and interesting, to details the molecular pathways of EMT and cytokine/chemokines secretion, in function of the embryologic derivation of the tissue in which the tumor arises. Finally, all NCDTs retain a distinctive trait related to the common upregulation of HIF-dependent pathway and relative hypoxia-response element (HRE). In particular, overexpression of HIF was reported in pheochromocytoma, neuroblastoma, meningioma and in cardiac myxoma and increased expression of HRE in melanoma and shawannomas, small cell lung cancer and in medullary thyroid carcinoma was reported. HIF and HER expressions are strictly connected to the degree of intratumoral hypoxia. Effects of hypoxia and HIF on the early immune response differ depending on local stimuli, and the level and duration of HIF activation, determined by tissue oxygenation and possibly through signaling cascades activated downstream. HIF-1 is crucial for macrophage-mediated inhibition of T cells in hypoxic conditions [119]. In hypoxic areas of tumors, the expression of matrix metalloproteinase proteins (MTX) cleave the Fas ligand from neighboring cells, making tumor cells less responsive to lysis by natural killer (NK) and T cells [120]. The cis-element of the promotor gene of MTX allowing the regulation by a various set of trans-activators including hypoxia-elements response (HER) such as NFKB (https://onlinelibrary.wiley.com/doi/pdf/10.1002/jcp.20948). On this basis, the known role of HIF during the development of the neural crest, provokes a reflection on how this pathway could represent connection and an interposition point respectively with the carcinogenesis and progression of NCDTs. The points of connection concern the trigger mechanism of the epithelial–mesenchymal transition contemporary with the inhibition of the adaptive immune response by a reactivation of the constitutive HIF pathway. The interposition point is linked to the additive hypoxia increment, proportional to the tumor volume, which sustaining the constitutive expression of HIF and HER implements the immunosuppressive microenvironment surrounding NCDTs. Interestingly, it has been established that HIF is involved in the regulation of PD-L1. HIF-1α regulates PD-L1 by binding to a hypoxia response element (HRE) of the PD-L1 promoter and activates PD-L1 transcription. Consistently, increased HIF-1 levels are associated with increased PD-L1 expression and that can induce inhibition of T-cell functions. In such a way hypoxia can also result in immune suppression [121].

## Figures and Tables

**Figure 1 cancers-12-00840-f001:**
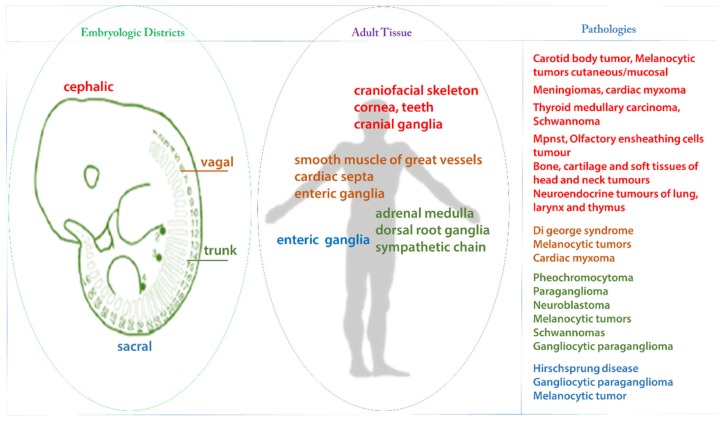
Segments of neural crest with relative adult tissues and pathologies derivate.

**Figure 2 cancers-12-00840-f002:**
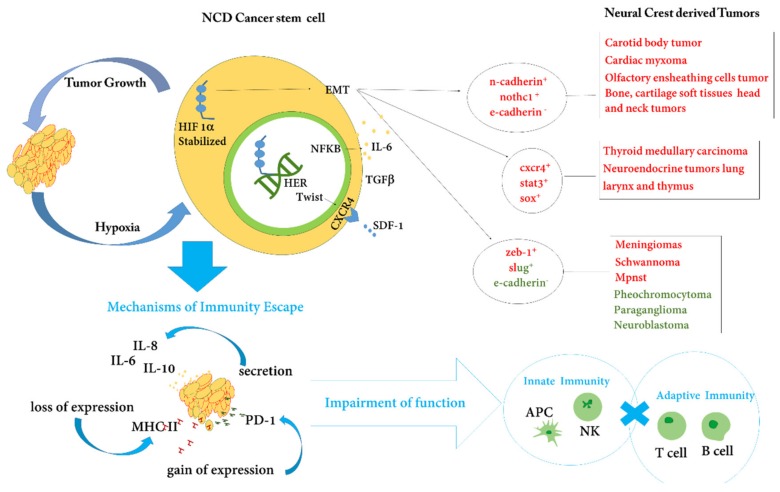
Stem cells of the neural crest-derived tumors (NCDTs) make epithelial–mesenchymal transition (EMT) pathway reactivating together with the immunological suppression strategies useful during the embryonic life to guarantee maternal-fetal and intra-embryonic immune tolerance. The principal antigens connected to EMT in NCDTs are detailed related to the type of tumor. Moreover, NCDTs maintain a distinctive trait related to the common upregulation of the HIF (hypoxia-inducible factor) -dependent pathway. Hypoxia-inducible factor (HIF) and hypoxia response elements (HRE) are used by embryonic stem cells to modulate their own stemness. In the tumor context, HIF triggers EMT in cancer cells, both HIF and EMT favor cytokine release inducing inflammation. These actions are sustained by hypoxia, that during tumor growing increased proportional to the tumor volume, and induces HER HIF-dependent. HIF determine a dependent positive feedback loop that included NF-kB, IL-6, STAT3, favoring further cytokines are released to trigger inflammation and to impair innate immunity reactions. Both TWIST1 and TGF-β1 are, in turn, regulated by directly or indirectly binding to HER HIF-dependent. TWIST1 is able to repress expression of E-cadherin, to coordinate EMT and chemotaxis toward the chemokine SDF-1 by upregulating its receptor CXCR4. SDF1/CXCR4 is involved in the chemotactic guidance and its activation is common in the progression phase of NCDTs. Finally, both EMT and HIF pathways in NCDTs are able to induce MHC II loss of expression and gain of expression of PD-1, impairing the adaptive immunity by B and T cells function failure.

**Table 1 cancers-12-00840-t001:** Neural crest-derived cell types and tumors.

Embryonic Derivation	Neural Crest-Derived Cell Types				Related Tumors
	Neurons and Glial Cells	Pigment Cells	Endocrine Cells	Mesenchymal Cells	
**Cephalic**	Sensory cranial ganglionic neurons Parasympathetic (ciliary)ganglionic neuron				Paraganglioma
		Skin melanocytes, Choroid melanocytes, Other extracutaneous melanocytes (in gums, meninges, etc.)			Melanocytic tumors
	Schwann cells along PNS nerves				Schwannoma MPNST
	Olfactory Ensheating Cells				Olfactory ensheating cells tumors
			Calcitonin-producing cells		Thyroid medullary carcinoma
			Carotid body cells		Carotid body tumor
				Meninges (forebrain)	Meningiomas
				Craniofacial Osteocytes and Chondrocytes, Adipocytes, Dermal cells	Bone, cartilage and soft tissues of head and neck
	Olfactory neuroepithelium				Esthesioneuroblastoma
			Neuroendocrine cells in larynx, lungs, and thymus		Neuroendocrine tumors larynx, lungs, and thymus
Vagal	Enteric ganglionic neurons	Melanocytes			Melanocytic tumors
				Vascular smooth muscle cells Heart conotruncus	Cardiac Myxoma
Trunk Enteric ganglionic neurons	Sympathetic ganglionic neurons Parasympathetic ganglionic neurons		Adrenal medullary cells		Pheochromocitoma Paraganglioma Neuroblastoma
		Melanocytes			Melanocytic tumors
	Schwann cells along PNS nerves				Schwannoma
	Sensory (dorsal root) ganglionic neurons Satellite glial cells in PNS ganglia				Peripheral Gangliocytoma
	Enteric ganglionic neurons				Gangliocytic Paraganglioma
Sacral		Melanocytes			Melanocytic tumors
	Enteric ganglionic neurons				Gangliocytic paraganglioma

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
