# Peer review of "Innate and Adaptive Immunity Linked to Recognition of Antigens Shared by Neural Crest-Derived Tumors"

_cancers, 2020, doi:10.3390/cancers12040840_

Round 1
Reviewer 1 Report
The paper by Donato, G. is quite well-thought, thought-provoking and properly designed. I only have a problem related to the professionalism and length of the paper.
(1) It would be easier for the people not related to the field and even the people directly working on the field to check the paper with a shorter length.
(2) The professionalism of the paper can also be improved by improving the quality of the table and figures.
If the authors can take care of these two issues, the paper can be a valuable contribution to the field.
Author Response
Review 1:
The paper by Donato, G. is quite well-thought, thought-provoking and properly designed. I only have a problem related to the professionalism and length of the paper.
(1) It would be easier for the people not related to the field and even the people directly working on the field to check the paper with a shorter length.
(2) The professionalism of the paper can also be improved by improving the quality of the table and figures.
If the authors can take care of these two issues, the paper can be a valuable contribution to the field.
Thanks for the appreciation. As regards the brevity of the manuscript, we have tried to streamline it as much as possible, trying to realize that we have also added text in relation to the requests of the second reviewer. The figures and the table have been modified and we hope that in this graphic look they will meet your requests.
Reviewer 2 Report
Donato et al present a review article on the role of immune evasion of NCDT by expression of specific embryo antigens. The focus is on the role of EMT-mediated dissemination of those tumor cells and control of immune microenvironment. Although, overall this article is interesting there are several issues related to the presentation and cohesiveness of this work. Abstract should be re-written to better present the content of the article. Since this is a special a special Issue of Cancers on: "Targeting Innate Immunity Cells in Cancer", authors should make it more relevant to this topic.
Instead of presenting coherent review of the links between immune responses and EMT, authors provide a list of EMT genes/proteins that are affected and what their function is in 2.1. It would be more interesting to discuss those genes in context of immune evasion by NCDT. More details on the role of SDF-1/CXCR4 pathways in context of immune function should be provided. Authors do not describe any functional links between inflammation and NCDT but rather association studies. It would be more beneficial to present the topic as a coherent review of comparing different characteristics of NCDT in context of immune evasion, rather than describing each of the tumor type separately without commenting how those are linked and how we can use this for therapeutic benefit.
There are multiple grammatical errors and poorly written sections, which distract from the main theme of this review. There are also multiple colloquial statements: ‘ actin phenotype’, ‘cleavage of epithelial cadherin resolves into release of C-terminal fragments able to induce’, ect. – those should be replaced and the text should be re-written for better presentation.
Author Response
Review 2:
Donato et al present a review article on the role of immune evasion of NCDT by expression of specific embryo antigens. The focus is on the role of EMT-mediated dissemination of those tumor cells and control of immune microenvironment. Although, overall this article is interesting there are several issues related to the presentation and cohesiveness of this work. Abstract should be re-written to better present the content of the article. Since this is a special a special Issue of Cancers on: "Targeting Innate Immunity Cells in Cancer", authors should make it more relevant to this topic.
We have re-written the abstract to better present the content of the article
Instead of presenting coherent review of the links between immune responses and EMT, authors provide a list of EMT genes/proteins that are affected and what their function is in 2.1. It would be more interesting to discuss those genes in context of immune evasion by NCDT.
We have discuss EMT genes in context of immune evasion by NCDT as you suggest and the modifications are reported in paragraph 2.1
More details on the role of SDF-1/CXCR4 pathways in context of immune function should be provided.
Details on the role of SDF-1/CXCR4 pathways are insert in the paragraph 2.1, 2.2. and 3.2
Authors do not describe any functional links between inflammation and NCDT but rather association studies. It would be more beneficial to present the topic as a coherent review of comparing different characteristics of NCDT in context of immune evasion, rather than describing each of the tumor type separately without commenting how those are linked and how we can use this for therapeutic benefit.
Functional links between inflammation and NCDT are discuss as you suggest and the modifications are reported in conclusions and in figure 2
There are multiple grammatical errors and poorly written sections, which distract from the main theme of this review. There are also multiple colloquial statements: ‘ actin phenotype’, ‘cleavage of epithelial cadherin resolves into release of C-terminal fragments able to induce’, ect. – those should be replaced and the text should be re-written for better presentation.
We have corrected typos everywhere in the MS and we improved English and style of the MS. Where necessary, we have rewritten words or sentences to convey more clearly the message of the paper.

Round 2
Reviewer 2 Report
no further comments.